# Inverse Rational Control with Partially Observable Continuous Nonlinear Dynamics

**Minhae Kwon**
School of Electronic Engineering
Soongsil University
Seoul, Republic of Korea
minhae@ssu.ac.kr

**Saurabh Daptardar**
Google Inc.
Mountain View, CA, USA
saurabh.dptdr@gmail.com

**Paul Schrater**
Department of Computer Science
University of Minnesota
Minnesota, IN, USA
schrater@umn.edu

**Xaq Pitkow**
Electrical and Computer Engineering
Rice University
Houston, TX, USA
xaq@rice.edu

## Abstract

A fundamental question in neuroscience is how the brain creates an internal model of the world to guide actions using sequences of ambiguous sensory information. This is naturally formulated as a reinforcement learning problem under partial observations, where an agent must estimate relevant latent variables in the world from its evidence, anticipate possible future states, and choose actions that optimize total expected reward. This problem can be solved by control theory, which allows us to find the optimal actions for a given system dynamics and objective function. However, animals often appear to behave suboptimally. Why? We hypothesize that animals have their own flawed internal model of the world, and choose actions with the highest expected subjective reward according to that flawed model. We describe this behavior as *rational* but not optimal. The problem of Inverse Rational Control (IRC) aims to identify which internal model would best explain an agent's actions. Our contribution here generalizes past work on Inverse Rational Control which solved this problem for discrete control in partially observable Markov decision processes. Here we accommodate continuous nonlinear dynamics and continuous actions, and impute sensory observations corrupted by unknown noise that is private to the animal. We first build an optimal Bayesian agent that learns an optimal policy generalized over the entire model space of dynamics and subjective rewards using deep reinforcement learning. Crucially, this allows us to compute a likelihood over models for experimentally observable action trajectories acquired from a suboptimal agent. We then find the model parameters that maximize the likelihood using gradient ascent. Our method successfully recovers the true model of rational agents. This approach provides a foundation for interpreting the behavioral and neural dynamics of animal brains during complex tasks.

## 1 Introduction

Brains evolved to understand, interpret, and act upon the physical world. To thrive and reproduce in a harsh and dynamic natural environment, brains, therefore, evolved flexible, robust controllers. To be the controller, the fundamental function of the brain is to organize sensory data into an internal model of the outside world. The animals are never able to get complete information about the world. Instead,

they only get partial and noisy observations of it. Thus, the brain should build its own internal model which necessarily includes uncertainties of the outside world, and base its decision upon that model [1]. However, we hypothesize that this internal model is not always correct, but the animals still behave rationally — meaning that animals act optimally according to their own internal model of the world, which may differ from the true world.

The goal of this paper is to identify the internal model of the agent by observing its actions. Unlike Inverse Reinforcement Learning (IRL) [2, 3, 4] which aims to learn only the reward function of target agent, or Inverse Optimal Control (IOC) [5, 6] to infer only unknown dynamics model, we use Inverse Rational Control (IRC) [7] to infer both. Since we consider neuroscience tasks which include naturalistic controls and complex physics of the world, we substantially extend past work [7] to include continuous spaces of state, action, and parameter with nonlinear dynamics. We parameterize nonlinear task dynamics and reward functions based on a physics model such that the family of tasks shares an overall structure but has different model parameters. In our framework, an *experimentalist* can observe state information of the environment and actions taken by the agent. On the other hand, the experimentalist cannot observe information about the agent's internal model, such as its observations and beliefs. IRC infers the latent internal information of the agent using the data observable by the experimentalist.

The task is formulated as a Partially Observable Markov Decision Process (POMDP) [8, 9], a powerful framework for modeling agent behavior under uncertainty. In order to model an animal's cognitive process whereby the decision-making is based on its own beliefs about the world, we reformulate the POMDP as a belief Markov Decision Process (belief MDP) [10, 11]. The agent builds its belief (*i.e.*, its posterior distribution over world states) based on partial, noisy observations and its internal model, and the decision-making is based on its belief.

We construct a Bayesian agent to learn optimal policies and value functions over an entire parameterized family of models, which can be viewed as an optimized ensemble of agents each dedicated to one task. This then allows us to maximize the likelihood of the state-action trajectories generated by a target agent, by finding which parameters from the ensemble best explain the target agent's data.

The main contributions of this paper are the following. First, our work is able to find both the reward function and internal dynamics model simultaneously in continuous nonlinear tasks. Note that continuous nonlinear dynamical systems are the most general form of tasks, so it is trivial to solve discrete and/or linear systems using the proposed approach. Second, we propose a novel approach to implement the Bayesian optimal control ensembles, including an idea of belief representation and belief updating method using estimators with constrained representational capacity (*e.g.*, an extended Kalman filter). This allows us to build an algorithm that imitates the bounded rational cognitive process of the brain [12] and to perform belief-based decision-making. Lastly, we propose a novel approach to IRC combining Maximum Likelihood Estimation (MLE) and Monte Carlo Expectation-Maximization (MCEM). This method successfully infers the reward function and internal model parameters of the target agent by maximizing the likelihood of state-action trajectories under the assumption of rationality, while marginalizing over latent sensory observations. Importantly, this is possible because we trained ensembles of agents over entire parameter spaces using flexible function approximators. To the best of our knowledge, our work is the first study to infer both the reward and internal model of an unknown agent with partially observable continuous nonlinear dynamics.

## 2   Related Work

**Inverse reinforcement learning (IRL).** The goal of IRL or imitation learning is to learn a reward function or a policy from expert demonstrations, and the goal of Inverse Optimal Control (IOC) is to infer an unknown dynamics model. Both approaches solve aspects of the general problem of inferring internal models of an observed agent. For example, some IRL works such as [13, 14, 15, 16] formulate the optimization problems to find features of reward or cost function that best explain the target agent's state-action trajectories. Specifically, [13] finds reward features by solving a linear programming problem, and [15] uses a quadratic programming method to learn mappings from features to a cost function. In addition, [17] combines the principle of maximum entropy [18] to IRL so that the solution becomes as random as possible while still matching reward features the best. This guarantees avoiding the worst-case policy [19, 20]. Another stream of IRL is imitation learning [21, 22, 23, 24]. Typical IRL approaches use a two-step process: first learn the expert's

reward function first, and then train the policy with the learned reward. This could be slow, [21] directly extracts a policy from data. Across all of these methods, there is no a complete inverse solution that can learn how an agent models rewards, dynamics, and uncertainty in a partially observable task with continuous nonlinear dynamics and continuous controls.

**Meta reinforcement learning (Meta RL).** The fundamental objective of Meta RL is to learn new skills or adapt to a new environment rapidly with a few training experiences. In order to efficiently adapt to the new tasks or environments, some Meta RL works try to infer tasks or meta parameters that govern the general task. For example, optimization-based meta RL works such as [25, 26, 27, 28] include a so-called 'outer loop' which optimizes the meta-parameters. In this sense, the meta RL is related to our goal since both work aim to infer the task parameters, although we use this parameterization to explain the actions of an agent. However, there are few studies to consider the partially observable setting of the agent. [29] includes both POMDP frameworks and meta-learning, but the partially observable information is the task information not the state information. [30] also considers a Bayesian approach with meta-learning, but it also uses Bayesian reasoning to infer the unseen tasks and learn quickly. Therefore, our paper differs from other Meta RL works in its task structure and goal. We allow partial observability about the world state, as occurs naturally in the animal's decision-making process. More fundamentally, the goal of our work is not to find smarter agents, but rather to infer the internal model of an existing agent and to explain its behaviors.

**Neuroscience and cognitive science** Neuroscientists aim to answer how the brain selects actions based on noisy sensory information and incomplete knowledge of the world. The hypothesis of the Bayesian brain [31] has been proposed to explain the brain's functionalities with Bayesian inference and probabilistic representations of the animal's internal state. Several studies propose mechanisms by which neurons could implement optimal behaviors despite pervasive uncertainty [11, 32, 33]. Despite the utility of having behavioral benchmarks based on optimality, animals often appear to behave suboptimally. Such suboptimality might come from the wrong internal model [7, 34] that is induced by a subjective prior belief of the animal [35, 36, 37], bounded rationality [38, 39, 40] and suboptimal inference [41]. The main goal of this paper is to infer the internal model of suboptimal agents using state-of-the-art deep reinforcement learning techniques and to provide a theoretical tool to interpret behavior and neural data obtained from ongoing neuroscience research.

For this reason, we test our approach by simulating an existing neuroscience task called 'catching fireflies' [42, 43], which is complex enough to require a sophisticated internal model, while being restrictive enough that animals can learn it and one can adequately constrain models of this behavior using feasible data volumes. Ultimately we will apply our approach to understand the internal models of behaving animals, where we do not know the ground truth. Before doing that, it is important to use simulated agents that allow us to validate the method when we do know the ground truth. Recently, a similar effort to build AI-relevant testbed for animal cognition and behavior is presented in [44, 45].

## 3 Bayesian Optimal Control Ensembles

Our method can be viewed as a search over an ensemble of agents, each optimally trained on different POMDP tasks, to find which of these agents best explain the experimentally observed behaviors of a target agent. The experimentalist is an external observer who has information about the world states and agent actions, but not about the agent's internal model, noisy sensory observations, or beliefs.

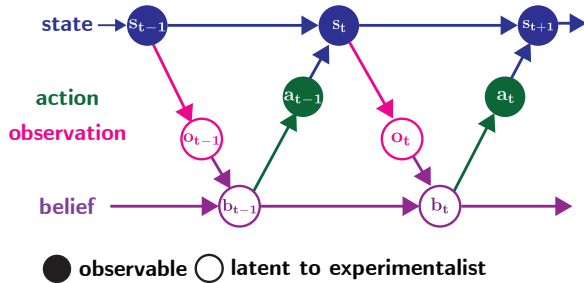

Figure 1: Graphical model of a POMDP. Solid circles denote observable variables to an experimentalist, and empty circles denote latent variables.

### 3.1 Belief Markov Decision Process and optimal control

A POMDP is defined as a tuple $M = (\mathcal{S}, \mathcal{A}, \Omega, R, T, O, \gamma)$ that includes states $s_t \in \mathcal{S}$, actions $a_t \in \mathcal{A}$, observations $o_t \in \Omega$, reward functions $R(s_t, a_t, s_{t+1}; \theta)$, state transition probabilities $T(s_{t+1}|s_t, a_t; \theta)$, observation probabilities $O(o_t|s_t; \theta)$ at time $t$, and a temporal discount factor $\gamma$. Here, $\theta \in \Theta$ denotes a vector of model parameters defining the rewards, state transitions and

observations, and the state space $\mathcal{S}$ and action space $\mathcal{A}$ are considered to occupy a continuous space. Thus, $\theta$ parameterizes a POMDP family. A graphical model of a POMDP is presented in Figure 1.

The state $s_t$ is defined as the representation of the environment which can live in high dimensional spaces. It may be fully accessible by the experimentalist but not by the agent. The agent gets an observation $o_t$ of the environment with state $s_t$, which is partial and noisy version of state $s_t$. Because of the partial observability, the dimension of $o_t$ could be lower than the dimension of $s_t$. The observation process is modeled by the observation function $O(o_t|s_t; \theta)$. Note that any noise added from state to observation is the internal noise of the agent, *i.e.*, the noise within the nervous system of the agent. Because of this noise, the observation is only known to the agent and the experimentalist can never access it directly.

Based on its observations and actions up to time $t$, a rational agent builds a posterior distribution $B(s_t|o_{1:t}, a_{1:t-1}; \theta)$ over the world state given the history of observations and actions, and it bases its actions upon that posterior. In practice this posterior is summarized by a *belief* $b_t$, defined as sufficient statistics for the posterior distribution over states, *i.e.*, $B(s_t|b_t) = B(s_t|o_{1:t}, a_{1:t-1}; \theta)$. In principle, a belief $b_t$ over a general continuous state could be infinite-dimensional, but we assume that the belief is continuous but finite-dimensional. Let $B(s_t|b_t)$ be the probability that the environment is in the state $s_t$ when the agent's belief is $b_t$. By the Markov property, $b_t$ is determined by $b_{t-1}, a_{t-1}, o_t$ such that $B(s_t|b_t)$ can be calculated as follows.

$$B(s_t|b_t) = B(s_t|b_{t-1}, a_{t-1}, o_t; \theta) \tag{1}$$

$$= \frac{1}{Z} O(o_t|s_t; \theta) \int ds_{t-1} T(s_t|s_{t-1}, a_{t-1}; \theta) B(s_{t-1}|b_{t-1}) \tag{2}$$

where $Z = \int ds_t\, O(o_t|s_t; \theta) \int ds_{t-1} T(s_t|s_{t-1}, a_{t-1}; \theta) B(s_{t-1}|b_{t-1})$ is a normalizing constraint. In general this recursion is intractable, so we approximate it under tractable model assumptions, as we do in our application below. By replacing the state of the environment by the belief of the agent, the POMDP problem can be reformulated as a belief MDP problem, and the optimal policy can be found based on well-known MDP solvers [46, 47, 48, 49] applied to the fully observed belief state.

The optimal policy $\pi^*(a_t|b_t; \theta)$ defines how the agent chooses an action $a_t^*$ that maximizes the temporally discounted total expected future reward, given the current belief $b_t$ and internal model $\theta$. This defines the $Q$-value $Q(b_t, a_t; \theta)$ as a belief-action value:

$$Q(b_t, a_t; \theta) = \int db_{t+1} \overline{T}(b_{t+1}|b_t, a_t; \theta) \left( \overline{R}(b_t, a_t, b_{t+1}; \theta) + \gamma \max_a Q(b_{t+1}, a; \theta) \right) \tag{3}$$

where $\overline{T}(b_{t+1}|b_t, a_t; \theta)$ is the belief transition probability and $\overline{R}(b_t, a_t, b_{t+1}; \theta)$ is the reward as a function of belief, defined as follows.

$$\overline{T}(b_{t+1}|b_t, a_t; \theta) = \iiint ds_t\, ds_{t+1}\, do_{t+1}\, B(s_t|b_t) T(s_{t+1}|s_t, a_t; \theta) O(o_{t+1}|s_{t+1}; \theta) p(b_{t+1}|b_t, a_t, o_{t+1}; \theta)$$
$$\tag{4}$$

$$\overline{R}(b_t, a_t, b_{t+1}; \theta) = \iint ds_t\, ds_{t+1} B(s_t|b_t) B(s_{t+1}|b_{t+1}) R(s_t, a_t, s_{t+1}; \theta)$$

In (4), the belief update is expressed in a generalized form $p(b_{t+1}|b_t, a_t, o_{t+1}; \theta)$ that allows either deterministic optimal belief updates, or could even account for other constraints on the inference process, including stochasticity.

The optimal action from a belief state will be also defined by a deterministic policy $\pi^*(a_t|b_t; \theta) = \delta(a_t^* = \arg\max_a Q(b_t, a; \theta))$. In case of continuous belief and action spaces, it is hard both to compute an optimal $Q$-function and to maximize it. Thus, we will approximate both using neural networks.

### 3.2 Training Bayesian optimal control ensembles with partial noisy observations

To successfully design and train an ensemble of agents, we identify three major challenges and provide solutions.

First, how can we construct the *optimal control ensembles* that can solve a family of tasks? As discussed, the task can be parameterized by the model parameter $\theta \in \Theta$ such that the family of tasks

shares the model structure but has different model parameters. We use this model parameter as an external input to flexible function approximators (neural networks) to estimate values and policies (Critic and Actor). Thus, the agent can be trained over parameter spaces. As presented in Figure 2, Critic and Actor both take parameter vector $\theta$ as an input, and respectively calculate the $Q$-value and best action for the task with that $\theta$.

Second, how should we represent and update the agent's belief? For our concrete example application below, we use an extended Kalman filter [50] to provide a tractable Gaussian approximation for the belief state and its nonlinear dynamics. The resultant belief update is deterministic, $p(b_{t+1}|b_t, a_t, o_{t+1}; \theta) = \delta\left(b_{t+1} = f(b_t, a_t, o_{t+1}; \theta)\right)$. Tests with more flexible particle filters showed that this approximation is reasonable in our target application. For other applications, different belief representations and dynamics may be more accurate [51], and in principle a family of agents could use representational learning [52].

Lastly, how should we train a rational model agent ensemble with continuous belief and action spaces? Here we use the model-free deep reinforcement learning algorithm called Deep Deterministic Policy Gradient (DDPG) [53]. This method is able to approximate the value function over continuous belief states, actions, and task parameters, all using one neural network (the Critic), and uses it to train a policy network (the Actor) which also receives inputs about the current belief and task parameters. Viable alternatives for continuous control in the deep reinforcement learning literature include [54, 55, 56, 57].

The training process for optimal control ensembles is summarized in Algorithm 1, and a block diagram is provided in Figure 2. The agent is trained on simulated experiences. Given the belief $b_t$ and parameters $\theta$, the Actor returns the best action $a_t$. As the agent performs the action $a_t$, it changes the world state to $s_{t+1}$ following the state transition probabilities $T$. The reward from the world $R$ is given to the agent and fed back to the Critic to get a better estimation of the $Q$-value, which then improves the selection of the action in the Actor. From the new state $s_{t+1}$, the agent gets a partial and noisy observation $o_{t+1}$ with the observation probabilities $O$. Then, the Gaussian belief state is updated using the extended Kalman filter $f$, $b_{t+1} = f(b_t, a_t, o_{t+1}; \theta)$. A new action $a_{t+1}$ is selected by the Actor, and these processes are iterated until the neural networks are fully trained. During this training, we sample new model parameters $\theta$ every episode so the agent can experience the entire space of tasks, and thus generalize better over that space.

---

**Algorithm 1:** Train Bayesian optimal control ensembles

**Initialization:** Initialize Actor and Critic
**repeat**
    t = 0, Reset $s_0, b_0$
    Sample model parameter $\theta \sim$ prior $\mathcal{P}(\Theta)$
    **repeat**
        Select action $a_t \leftarrow \text{Actor}(b_t; \theta)$
        Sample new state
         $s_{t+1} \sim T(s_{t+1}|s_t, a_t; \theta)$
        Get reward $r = R(s_t, a_t, s_{t+1}; \theta)$
        Train Critic by back-propagating $r$
        Calculate $Q$-value $q \leftarrow \text{Critic}(b_t, a_t; \theta)$
        Train Actor by back-propagating $q$
        Sample new observation
         $o_{t+1} \sim O(o_{t+1}|s_{t+1}; \theta)$
        Update belief $b_{t+1} \leftarrow f(b_t, a_t, o_{t+1}; \theta)$
         using the extended Kalman filter
        $t \leftarrow t + 1$
    **until** *episode ends*;
**until** *Actor and Critic are fully trained*;

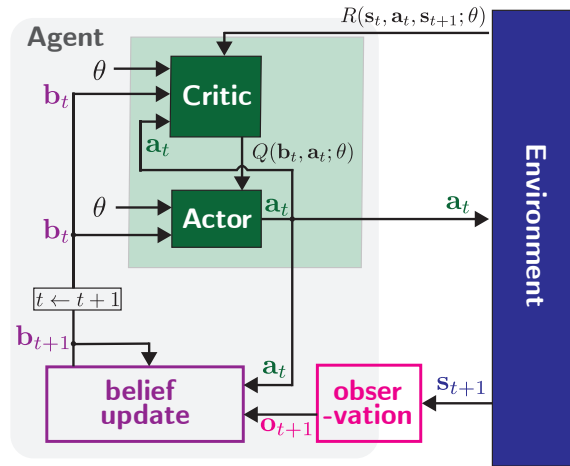

Figure 2: A block diagram of Algorithm 1.

## 4 Inverse Rational Control with Maximum Likelihood Estimation

Once an agent ensemble is fully trained over the entire parameter space, we can use this ensemble to find the internal model parameters of the best-fitting rational agent in that model family. We solve

the continuous Inverse Rational Control problem by finding the parameters $\theta$ that have the highest likelihood for explaining an agent's measured behavior.

## 4.1 Discrepancy between the true world and internal model

Recall that our core hypothesis is that animals have their own internal model of the world which may not be always correct, but they still behave *rationally*, choosing actions with the highest expected subjective reward according to their internal model. We must therefore distinguish between the two kinds of model parameters: the true ones $\phi$ which determine the world dynamics and are known to the experimentalist, and the agent's internal model parameters $\theta$ which are latent for the experimentalist but governs all cognitive processes of the agent (Figure 3). The world parameters $\phi$ govern the world dynamics such as state transition probability $T(s_{t+1}|s_t, a_t; \phi)$ and reward function $R(s_t, a_t, s_{t+1}; \phi)$. On the other hand, the internal parameters $\theta$ govern the agent's internal process such as the observation probability $O(o_t|s_t; \theta)$, the belief transition probability $\overline{T}(b_{t+1}|b_t, a_t; \theta)$, and the subjective reward as a function of belief $\overline{R}(b_t, a_t, b_{t+1}; \theta)$, leading to a subjective belief update probability $p(b_{t+1}|b_t, a_t, o_{t+1}; \theta)$ and rational policy $\pi(a_t|b_t; \theta)$.

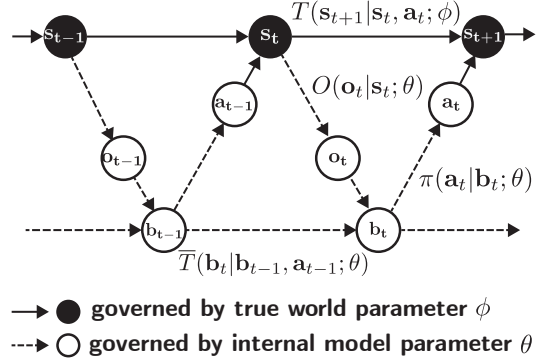

→ ● governed by true world parameter $\phi$
--→ ○ governed by internal model parameter $\theta$

Figure 3: An illustrative explanation of model discrepancy. The solid lines and circles are governed by the true world parameter $\phi$ which is known to the experimentalist. The dashed lines and empty circles are governed by internal model parameter $\theta$ which is latent to the experimentalist, and may differ from $\phi$.

## 4.2 Inferring internal model parameter $\theta$

To find the internal model parameters $\theta$ that maximize the log-likelihood of the experimentally observable data $(s, a)_{1:T}$, $\hat{\theta} = \arg\max_\theta \ln p(s_{1:T}, a_{1:T}|\phi, \theta)$ we use the Monte Carlo Expectation Maximization (MCEM) algorithm [58] to marginalize the complete data log-likelihood over latent observations $o_{1:T}$ and beliefs $b_{1:T}$. This yields an iterative algorithm, which repeatedly maximizes

$$\hat{\theta}_{k+1} = \arg\max_\theta \int do_{1:T}\, db_{1:T}\, p(o_{1:T}\, b_{1:T}|s_{1:T}, a_{1:T}; \theta_k) \ln p(s_{1:T}, o_{1:T}, b_{1:T}, a_{1:T}|\phi, \theta) \quad (5)$$

$$\approx \arg\max_\theta \frac{1}{L} \sum_{l=1}^{L} \ln p(s_{1:T}, o_{1:T}^{(l)}, b_{1:T}^{(l)}, a_{1:T}|\phi, \theta) \quad (6)$$

where the sum is over samples $(o^{(l)}, b^{(l)})_{1:T}$ drawn from a posterior distribution $p(o_{1:T}\, db_{1:T}|s_{1:T}, a_{1:T}; \theta_k)$ determined by parameters $\theta_k$ from previous iterations.

The log-likelihood of the complete data (including the $l$-th samples of observations and beliefs based on parameter $\theta_k$) can be decomposed using the Markov property into

$$\ln p(s_{1:T}, o_{1:T}^{(l)}, b_{1:T}^{(l)}, a_{1:T}|\phi, \theta) \quad (7)$$

$$= \ln p(s_0, o_0^{(l)}, b_0^{(l)}, a_0) + \sum_{t=1}^{T} \Big( \ln T(s_t|s_{t-1}, a_{t-1}; \phi) + \ln O(o_t^{(l)}|s_t; \theta)$$

$$+ \ln p(b_t^{(l)}|b_{t-1}^{(l)}, a_{t-1}, o_t^{(l)}; \theta) + \ln \pi(a_t|b_t^{(l)}; \theta) \Big). \quad (8)$$

Note that the only terms depending on the agent's parameters $\theta$ are the latent observations probabilities, belief dynamics, and policy. So when we optimize over $\theta$, all other terms vanish. Moreover, since we use deterministic belief updates, the belief update term in (8) is also independent of $\theta$ when evaluated on sampled beliefs. The only terms that survive are

$$\hat{\theta} = \arg\max_\theta \sum_{l=1}^{L} \sum_{t=1}^{T} \Big( \ln O(o_t^{(l)}|s_t; \theta) + \ln \pi(a_t|b_t^{(l)}; \theta) \Big). \quad (9)$$

To optimize (9), we use gradient ascent over parameter space, $\theta \leftarrow \theta + \alpha \nabla_\theta \mathcal{L}$ with learning rate $\alpha$. This is explained in Algorithm 2.

---

**Algorithm 2:** Estimate $\theta$ that explains externally observable data the best

---

**Data:** Collected data by the experimentalist: $s_{0:T}, a_{0:T}$
$T$: the length of a trajectory
$L$: the number of samples
$b_0 = \mathcal{N}(o_0, 10^{-6})$
**Initialization:** Initialize $\theta$ with a random sample from the prior $\theta \sim \mathcal{P}(\Theta)$
**repeat**
    $\mathcal{L}(\theta) = 0$
    **for** $l = 1 : L$ **do**
        **for** $t = 1 : T$ **do**
            Sample $o_t^{(l)} \sim O(o_t^{(l)}|s_t; \theta)$
            Belief update using extended Kalman filter $b_t^{(l)} \leftarrow f(b_{t-1}^{(l)}, a_{t-1}, o_t^{(l)}; \theta)$
            $\mathcal{L}(\theta) \leftarrow \mathcal{L}(\theta) + \ln O(o_t^{(l)}|s_t; \theta) + \ln \pi(a_t|b_t^{(l)}; \theta)$
        **end**
    **end**
    Update $\theta$ using gradient ascent step: $\theta \leftarrow \theta + \alpha \bigtriangledown_\theta \mathcal{L}$
**until** $\mathcal{L}(\theta)$ *converges*;

---

## 5 Demonstration task: 'Catching fireflies'

To verify the proposed method, we carefully select a relevant task. Our application focus on continuous world states, actions, and beliefs makes standard RL testbeds (*e.g.* Nintendo, MuJoCo) more difficult. Common tasks like gridworld or tiger do not exhibit continuous properties and remain excessively small toy problems. Standard continuous control tasks do not use partially observability; tasks that do would likely generate beliefs that would be substantially harder to interpret. Additionally, there is a ready application to existing neuroscience experiments based on 'catching fireflies' in virtual reality [59, 43], which is complex enough to be interesting to animals, requires a continuous representation of uncertainty and continuous control, and yet remains tractable enough that we can assess the fidelity of recovered beliefs.

In our task, an agent must navigate through a virtual environment to reach a transiently visible target, called the 'firefly' (Figure 4A). At the beginning of each trial, a firefly blinks briefly at a random location on the ground plane. The agent is able to control its forward and angular velocities to freely navigate the world. If the agent stops sufficiently close to the invisible target, it receives a reward. As the agent moves, a sparse ground plane texture generates an optic flow pattern, a vector field of local image motion. This allows the agent to estimate its speed up to some perceptual uncertainty. However, there is no direct access to information about its current location because the ground plane texture is transient and does not provide spatial landmarks. Thus, the agent must integrate optic flow to estimate its current position relative to the firefly target, as well as its uncertainty about that position.

We demonstrate the efficacy of our approach using a simulated agent for which ground truth is known. Thus, we verify our method by showing the successful recovery of the internal model parameters since we know the ground truth. Note that there are no comparisons to alternative methods because no other algorithms exist that solve the IRC problem in continuous state and action spaces. Figure 4B shows a two-dimensional contour plot of the approximate log-likelihood of observable data $\mathcal{L}(\theta)$. Recall that the model parameters $\theta$ are high dimensional, so here we plot only two dimensions of $\theta$. The red line shows an example trajectory of parameters $\theta$ as IRC Algorithm 2 converges. Our approach estimates $\theta$ that maximizes the log-likelihood of the observable data $\mathcal{L}(\theta)$. Figure 4C shows that the estimated parameters recovered by our algorithm closely match the agent's true parameters. The hyperparameters used to produce the results are provided in Appendix B, and the relationship between the number of trajectories and the accuracy of the parameter recovery is discussed in Appendix C.

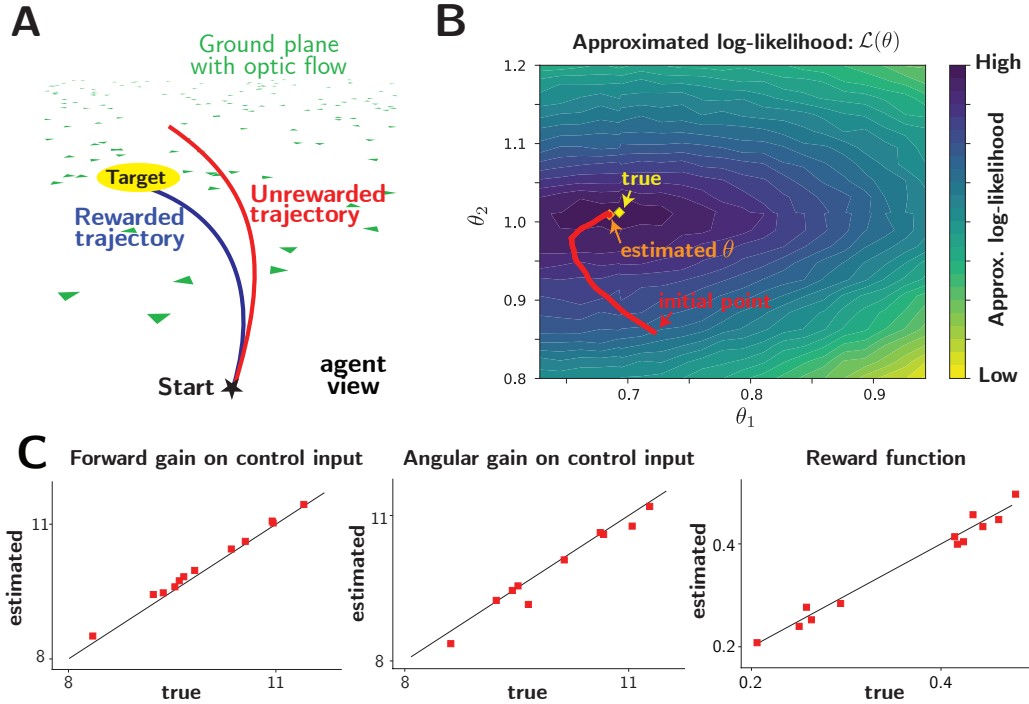

Figure 4: **A**. An illustration of the 'catching fireflies' task from the agent's point of view. To reach the transiently visible firefly target, an agent must navigate by optic flow over a dynamically textured ground plane. The agent is rewarded if it stops close enough to the target location. **B**. Converging trajectory of IRC estimates of the agent's parameters $\theta$. We use gradient ascent to find $\theta$ that maximizes approximated log likelihood $\mathcal{L}(\theta)$ in Algorithm 2. **C**. Successful recovery of individual agent parameters. The black line is the identity, meaning that true values and estimated values are equal. Across all parameter spaces, the proposed approach accurately recovers the agent's internal model parameters given limited data.

## 6 Conclusion

This paper introduces a novel framework to infer the internal model of agents from their behaviors. We infer not only the subjective reward function of the agent, but we also simultaneously infer the task dynamics that the agent assumes. To accomplish this, we first train Bayesian optimal control ensembles that generalize over the space of task parameters. Since the target agent is only exposed to partial information about the world state, the agent chooses the best action based on its belief about the world and its assumptions about the task. By using this optimally trained agent ensemble, our approach to Inverse Rational Control with continuous state and action spaces can infer the internal model parameters that best explain the collected behavioral data. With a simulated agent where we know the ground truth, we confirm that our approach successfully recovers the internal models. This success encourages us to apply this method to real behavioral data as well as to new tasks and applications.

## Broader Impact

We have implemented IRC for neuroscience applications, but the core principles have value in other fields as well. We can view IRC as a form of Theory of Mind, whereby one agent (a neuroscientist) creates a model of another agent's mind (for a behaving animal). Theory of Mind is a prominent component of human social interactions, and imputing rational motivations to actions provides a useful description of how people think [60, 61, 62]. Using IRC methods to provide a better understanding of people's motivations could yield important insights for understanding and improving social and political interactions, as well as raising possible ethical concerns if used for manipulation. The design

of agents interacting with humans would also benefit from being able to attribute rational strategies to others. For example, recent work uses a related approach to impute purpose to a neural network [16]. One important practical example is self-driving cars, which currently struggle to handle the perceived unpredictability of humans. While humans do indeed behave unpredictably, some of this may stem from ignorance of the rational computation that drives their actions. The IRC provides a framework for interpreting agents, and serves as a valuable tool for greater understanding of unifying principles of control.

**Acknowledgments**

The authors thank Dora Angelaki, James Bridgewater, Kaushik Lakshminarasimhan, Baptiste Caziot, Zhengwei Wu, Rajkumar Raju, and Yizhou Chen for useful discussions. MK, SD, and XP were supported in part by an award from the McNair Foundation. SD and XP were supported in part by the Simons Collaboration on the Global Brain award 324143 and NSF 1450923 BRAIN 43092. MK and XP were supported in part by NSF CAREER Award IOS-1552868. MK was supported in part by National Research Foundation of Korea grant NRF-2020R1F1A1069182. PS and XP were supported in part by BRAIN Initiative grant NIH 5U01NS094368.

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
