[Supplementary Material]

# Appendix

## A   Derivation of Monte Carlo Expectation Maximization (MCEM)

A derivation from (5) to (6) is based on MCEM. We here provide more details of the MCEM.

Let $x$ be the observable data, $z$ be the latent variable and $\theta$ be the parameters that govern the process. The goal is to find $\theta$ that maximizes the log likelihood of the observable data.

$$\theta = \arg\max_\theta \ln p(x|\theta)$$

The log likelihood of the observable data can be reformulated as follows.

$$
\begin{aligned}
\ln p(x|\theta) &= \int dz\, q(z) \ln p(x|\theta) \\
&= \int dz\, q(z) \big[ \ln p(x,z|\theta) - \ln p(z|x,\theta) \big] \\
&= \int dz\, q(z) \big[ \ln p(x,z|\theta) - \ln q(z) + \ln q(z) - \ln p(z|x,\theta) \big] \\
&= \int dz\, q(z) \Big[ \ln \frac{p(x,z|\theta)}{q(z)} - \ln \frac{p(z|x,\theta)}{q(z)} \Big] \\
&= \int dz\, q(z) \ln \frac{p(x,z|\theta)}{q(z)} - \int dz\, q(z) \ln \frac{p(z|x,\theta)}{q(z)} \quad\quad (10) \\
&= \mathcal{L}(q,\theta) + KL(q||p) \quad\quad (11)
\end{aligned}
$$

Since KL divergence is always non-negative value, $\mathcal{L}(q,\theta)$ is the lower bound of $\ln p(x|\theta)$. The complete data log likelihood $\ln p(x,z|\theta)$ is easier to handle than the observed data log likelihood $\ln p(x|\theta)$. Thus, instead of maximizing $\ln p(x|\theta)$, we aim to maximize its lower bound $\mathcal{L}(q,\theta) = \int dz\, q(z) \ln \frac{p(x,z|\theta)}{q(z)}$.

### A.1   E-step

As $KL(q||p)$ gets smaller, we have a tighter lower bound. If $KL(q||p) = 0$, $\ln p(x|\theta) = \mathcal{L}(q,\theta)$. $KL(q||p) = 0$ is satisfied only if $q = p$. Thus $q(z) = p(z|x,\theta)$ from (10). This is the E-step of the EM algorithm [63]. Note that in this step, $q(z)$ is a function only of $z$, which means both $x$ and $\theta$ are used as given variables. Thus, we denote $\theta_{\text{old}}$ as a fixed parameter that is used to specify $q(z)$. Once $q(z) = p(z|x,\theta_{\text{old}})$ is used in $\mathcal{L}(q,\theta)$ of (11), $\ln p(x|\theta)$ can be expressed as follows.

$$
\begin{aligned}
\ln p(x|\theta) &= \mathcal{L}(q,\theta) \\
&= \int dz\, p(z|x,\theta_{\text{old}}) \ln \frac{p(x,z|\theta)}{p(z|x,\theta_{\text{old}})} \\
&= \int dz\, p(z|x,\theta_{\text{old}}) \ln p(x,z|\theta) - \int dz\, p(z|x,\theta_{\text{old}}) \ln p(z|x,\theta_{\text{old}}) \\
&= \int dz\, p(z|x,\theta_{\text{old}}) \ln p(x,z|\theta) + H(z|x,\theta_{\text{old}}) \\
&= \mathcal{Q}(\theta,\theta_{\text{old}}) + H(z|x,\theta_{\text{old}}) \quad\quad (12)
\end{aligned}
$$

### A.2   M-step

Next, we want to find $\theta$ that maximizes $\ln p(x|\theta)$. This is the M-step of the EM algorithm. Since $H(z|x,\theta_{\text{old}})$ is a constant (i.e., not a function of $\theta$),

$$
\begin{aligned}
\theta &= \arg\max_\theta \ln p(x|\theta) \\
&= \arg\max_\theta \mathcal{Q}(\theta,\theta_{\text{old}}) \\
&= \arg\max_\theta \int dz\, p(z|x,\theta_{\text{old}}) \ln p(x,z|\theta). \quad\quad (13)
\end{aligned}
$$

If $p(z|x, \theta_{\text{old}})$ is hard to get analytically, (13) can be approximated by the Monte Carlo approach. The resultant optimization is called the MCEM algorithm.

$$\theta = \arg \max_{\theta} \int dz \, p(z|x, \theta_{\text{old}}) \ln p(x, z|\theta)$$

$$\approx \arg \max_{\theta} \frac{1}{L} \sum_{l=1}^{L} \ln p(x, z^{(l)}|\theta) \tag{14}$$

where $z^l$ is $l$-th particle for the latent variable $z$.

## B  Hyperparameters

| Algorithm 1 | | | |
|---:|---|---:|---|
| Batch size | 64 | Discount factor | 0.99 |
| Replay memory size | $10^6$ | Actor learning rate | $10^{-4}$ |
| Critic learning rate | $10^{-3}$ | Optimizer | Adam |
| Number of units per hidden layer | 128 | Activation function of hidden layer | ReLU |
| Activation function of Actor output layer | Tanh | Activation function of Critic output layer | Linear |
| Algorithm 2 | | | |
| Length of trajectory (T) | 500 | Number of samples (L) | 50 |
| Optimizer | Adam | Learning rate | $10^{-3}$ |

## C  Impact of The Number of Trajectories on Parameter Recovery

It is important to investigate the relationship between the number of data points and the accuracy of the parameter recovery to guide the experimentalists about how much data they need to collect for recovering a subject's internal model. The results presented in Figure 4 were from 500 state-action trajectories (500 fireflies), each with about 5–15 state-action time points. The amount of data is reasonable since the subjects repeat the task hundreds of times.

As one can easily expect, the recovery accuracy grows with data volume. Figure 5 explains the reason: the surface of log-likelihood becomes smoother and the peak moves closer to the agent's true parameters.

Figure 5: Log-likelihood surface with different numbers of data points. Red diamonds indicate true parameter values.