[Reviews · NeurIPS 2020]

Review 1

Summary and Contributions: The paper introduces a novel framework to infer the internal model (both reward function and dynamics).

Strengths: Their method is a novel in the sense that it is the first model to solve this problem. They combine several methods and apply them to this novel setting. The framework introduced is generalizable to many tasks. It’s a nice methodology paper. They demonstrate the effectiveness by recovering the internal model of the simulated agents. The code is shared, making it easy for other researchers to apply their method. I think it’s relevant to the board NeurIPS audience.

Weaknesses: It is clear that there’s no baseline model comparison as there exists no other model. The author also discusses the model can be directly applied to a simpler setting. It can be good to compare other models in a simpler setting, which also shows it can be generalized to other settings. --- Updates --- I appreciate the authors addressed my comments on sample efficiency. The results look good! Additional work with real dataset will make this paper much stronger and more impactful, which would be a nice plus for this paper. I'm keeping my score though I feel like this is a borderline paper without results on at least one real data. The model is aimed to provide a way to recover animal internal models, providing more insights into animal behavior. It would be interesting to apply the model to real dataset(s), to see what it can recover in addition to other existing findings.

Correctness: They are correct to the best of my knowledge.

Clarity: The paper is very well-written and I found it easy to follow.

Relation to Prior Work: The author extensively discussed some previous work and clearly pointed out how their work is different and novel compared to existing literature. I found their contribution is clear to me.

Reproducibility: Yes

Additional Feedback: I like this work, and I’m excited to see the new discovery using this method in animal/human behavior. I suppose to reliably recover the internal model, some minimum number of samples would be required, e.g. number of trials. It would be interesting to show a figure of how accurate the parameter recovery is as a function of the number of data points. It can also help experimentalists design their experiments properly if they are interested in recovering the internal model. There might be other constraining factors that experimentalist needs to take into account in order to use this model effectively, I would be interested to see a discussion.


Review 2

Summary and Contributions: The paper proposed inverse rational control (IRC), a framework and method for simultaneously inferring an agent's reward function (as in inverse RL and rational analysis) and internal model (as in inverse optimal control) under a limitation on the agent's belief computation modeled as an extended Kalman filter (as in cognitively-bounded rational analysis). The paper demonstrates this method on simulated data generated from an artificial agent operating in a simple neuroscience task.

Strengths: The overall ideas of a bounded agent (approximately) optimally maximizing an unknown reward under unknown constraints as a model for a behaving human / animal have a longstanding history in psychology/neuroscience/AI going back to Simon (as the paper reminds us). But combining all of the various possible limitations (model mismatch relative to the true world model, unknown reward, unknown dynamics) in a single framework is novel to my knowledge, in large part because this usually lands the researcher in a hard and under-constrained bilevel optimization problem with the lower level optimizing policy parameters w.r.t. reward and the upper optimizing fit parameters to data. The paper's proposal to address this issue by feeding the upper-level parameters as contextual variables into the lower-level policy optimization's state and thus solve the lower level problem only once (vs for every setting of the upper-level problem) is clever and elegant.

Weaknesses: The specific empirical evaluation chosen is the primary weakness of the paper. From a neuroscience perspective, the validation of parameter recovery on synthetic data is a necessary first step, but not a sufficient one. Given that [a] the task is primarily of neuroscientific interest and [b] a simpler (though also bayesian belief-updating) fit model is given in the cited prior work, the lack of comparison of cross-validated performance against that prior model is surprising. We should either see better cross-validation performance to the models in prior work, or similar performance but more insight / explanation of the underlying mental computation. This would show us a real payoff of the new insights here. Practically, I'm also concerned with whether this method is sample-efficient enough to address real neuroscience data, which tends to be small by modern ML standards. Outside of the neuroscience perspective, the fireflies task may be more interesting than canonical toy grid worlds, but is still far from realistic relative to application domains in robotics, self-driving cars, etc. == Post-Rebuttal Update == I have read through the other reviews and rebuttal. I appreciate the additional information about sample efficiency, and take the rebuttal's point that recovering the prior model would be more of a sanity check than a serious comparison because the prior model should be trivially recoverable in the current setup. At the same time, following discussion with the other reviewers I still think that the lack of empirical validation weakens the paper. I appreciate the difficulty of obtaining real neuroscientific data from humans or animals, but it doesn't change the fact that the neuroscience community is unlikely to engage with the work without empirical validation, and the RL contribution here is not significant by itself. Consequently, I am moving my rating up but I still think this is a borderline paper.

Correctness: Yes, as far as I can tell.

Clarity: Yes, and it was enjoyable to read.

Relation to Prior Work: For the most part, yes. Some closely related work on the psychology side more recent than Simon 1972 is missing (including from Anderson, Griffiths, Lewis, etc), but otherwise the connections across both the ML and neuroscience sides are discussed.

Reproducibility: No

Additional Feedback: For reproducibility: details are lacking on the specifics of the synthetic data experiment in terms of the synthetic agent parameters, as well as various details of the model parameters and hyperparameters -- there is not enough here to replicate from the paper or supplement. A github link to the code is provided in the supplement, which is good, but I did not click on it since I was concerned that it would break anonymity.


Review 3

Summary and Contributions: This paper considers the inverse rational control problem, which I summarize in my own words as follows: an RL agent interacts with a partially observable MDP using its believed optimal policy, which is induced by its own biased model prior, reward prior, hypothesis class, etc -- all of these internal (and therefore not directly observable) parameters are summarized notationally by theta. An experimentalist can only observe the action taken by the agent as well as the complete state and want to infer theta. The main contribution of the paper is three-fold: the authors first propose a way to learn an approximately policy in a POMDP using a belief based actor-critic variant; they then use MLE-like approach to inter theta, they finally demonstrate the effectiveness of their methodology on a real-world scenario. ___________________________________________________ Post-rebuttal update: I kept my score unchanged because I still think this paper is a intesesting combination of ideas. However, in order for this paper to be out of the borderline range of most reviewers, more serious experiments are definitely needed.

Strengths: I find the paper a blending of several interesting ideas aiming at a very interesting problem. While POMDP certainly has been studied for a long time, both theoretically and emprically, and the method proposed by the authors may be far from being the state-of-the-art, it is based on reasonable observations and intersting ideas, and is very suitable for the scenario they consider -- the inverse rational control problem. The problem itself may not have received a lot of attention in the RL community (and it is the first time I have ever heard of it), but for me it looks interesting enough and is a problem worth studying. There are in fact many non-trivial aspects of this problem: for example, how to characterize the belief, how to combine belief estimation and belief based value estimation into the same pipeline, etc., and the authors have addressed most of them.

Weaknesses: While I am buying the story, this paper certainly could use some improvement in the experiment section, especially if the authors want to get more attention from the RL community. Adding at least one standardized environment would most likely make the story more complete.

Correctness: Yes.

Clarity: Yes.

Relation to Prior Work: Yes.

Reproducibility: Yes

Additional Feedback:

[Author Response · NeurIPS 2020]

**Paper ID 3012: Inverse Rational Control with Partially Observable Continuous Nonlinear Dynamics** We would like to thank the reviewers for their valuable and important comments. We here submit our responses. All the issues addressed in this document will be included in the camera-ready version.

**1. Sample efficiency** [All reviewers] We agree it is important to investigate the relationship between the number of data points and the accuracy of the parameter recovery to guide the experimentalists about how much data they need to collect for recovering a subject's internal model. The results presented in the paper were from 500 state-action trajectories (500 fireflies), each with about 5–15 state-action time points. The amount of data is reasonable since the subjects repeat the task hundreds of times.

Figure 1: (Left) Recovery error vs the number of data points. (Right) Log-likelihood surface with different numbers of data points. Red diamonds indicate true parameter values.

We ran additional experiments to show the relationship between the number of state-action trajectories and the fractional error (absolute error divided by true parameter). We ran inverse rational control (IRC) for 20 agents with different model parameters, using 10 samples ($L$=10 in Algorithm 2) and 400 gradient ascent steps. Figure 1 (left) shows the performance given 10, 30 and 100 state-action trajectories. The error falls off with the square root of the number of trials as expected. These parameters can be readily identified with relatively few trials; for other tasks we expect the same scaling with numbers of trajectories, but with different scaling factors depending on the how much the actions vary with the task.

Figure 1 (right) explains why recovery accuracy grows with data volume: the surface of log-likelihood becomes smoother and the peak moves closer to the agent's true parameters.

**2. References in psychology** [R2] We thank the reviewer for suggesting related works in psychology. We have thoroughly reviewed the related works in psychology and added the following references with discussion in Section 2. Related works of the camera-ready version: Lieder et al., *Behavioral and Brain Sciences* (in press), Bourgin et al., *ICML* (2019), Krueger et al., *CogSci* (2018), Baker et al., *Nature Human Behavior* (2017) Rafferty et al., *Cognitive Science* (2015), Lewis et al., *Topics in cognitive science* (2014), Walsh et al., *Psychological Bulletin* (2014), Howes et al., *Psychological review* (2009).

**3. Hyperparameters for reproducibility** [R2] To increase reproducibility, we here provide details of hyperparameters for Algorithm 1 and Algorithm 2.

**4. Standard benchmarks** [R1,3] Now that we have a workable framework for IRC we do plan to apply it to new neuroscience and ML tasks (especially partially observed versions of standard continuous control tasks), since standard benchmarks for continuous POMDPs do not yet exist.

**5. Comparison to previous models** [R2] We do hope to access the behavioral data in [38,39] to directly compare our model to past work. However, our model essentially *contains* those models, which did not address control at all. So IRC will give identical findings when restricted to the older models but will provide fundamentally new explanations when including control.

| Algorithm 1 | | | |
|---|---|---|---|
| Batch size | 64 | Discount factor | 0.99 |
| Replay memory size | $10^6$ | Actor learning rate | $10^{-4}$ |
| Critic learning rate | $10^{-3}$ | Optimizer | Adam |
| Number of units per hidden layer | 128 | Activation function of hidden layer | ReLU |
| Activation function of Actor output layer | Tanh | Activation function of Critic output layer | Linear |
| Algorithm 2 | | | |
| Length of trajectory (T) | 500 | Number of samples (L) | 50 |
| Optimizer | Adam | Learning rate | $10^{-3}$ |

[Meta-Review · NeurIPS 2020]

The paper describes a novel technique for inverse rational control. The reviewers all agree that this is great work that makes an important contribution. There is one important weakness though: the experiments. More comprehensive experiments would be desirable to increase the impact of the work. Nevertheless, this is still good work.